# Tunable anharmonicity in cavity optomechanics in the unresolved sideband regime

**Jonathan L. Wise$^\star$, Clément Dutreix and Fabio Pistolesi$^\dagger$**

Université de Bordeaux, CNRS, LOMA, UMR 5798, F-33400 Talence, France

$\star$ jonathan.wise@u-bordeaux.fr,    $\dagger$ fabio.pistolesi@u-bordeaux.fr

## Abstract

Introducing a controlled and strong anharmonicity in mechanical systems is a present challenge of nanomechanics. In cavity optomechanics a mechanical oscillator may be made anharmonic by ponderomotively coupling its motion to the light field of a laser-driven cavity. In the regime where the mechanical resonating frequency and the single-photon coupling constant are small compared to the decay rate of the cavity field, it turns out that the quantum electromagnetic fluctuations of the laser field drive the oscillator into a high-temperature thermal state. The motional state may however be highly non-Gaussian; we show that a precise tuning of system parameters may even lead to a purely quartic effective potential for the mechanical oscillator. We present a theory that predicts the measurable signatures left by the mechanical anharmonicity. In particular, we obtain analytically and numerically the mechanical displacement spectrum, and explore the imprints of the mechanical anharmonicity on the cavity light field.

# 1   Introduction

Mechanical systems are emerging as promising candidates for performing tests of fundamental physical theories such as gravity (at high [1, 2] and low energy [3–6]), dark matter [7], and macroscopic quantum mechanics [8, 9]. In addition, the ability to generate and control nonclassical motional states opens the door to the development of many novel quantum technologies [10]. The particular focus on mechanical systems stems from the ease with which their motion may be coupled to other degrees of freedom. This may be understood simply by the fact that the mechanical motion of an object may change the boundary conditions imposed on other fields present. The resulting hybrid systems – known as electro-, magneto-, or optomechanical systems – have benefited from recent advances in drastically enhancing the isolation from the mechanical environment (e.g. Ref. [11]). However, one important ingredient often missing in these high finesse mechanical systems is anharmonicity, which may be a requirement for some of the exciting prospects mentioned above. While the small-amplitude motion of mechanical oscillators tends to be harmonic, anharmonicity may be induced via the coupling of the motion to another degree of freedom [12].

In cavity optomechanics, the motion of a mechanical element is coupled ponderomotively to the light field of a cavity due to radiation-pressure force [13, 14]. In such systems operating in the resolved sideband regime (also known as "good cavity"), where the oscillator motion is fast compared to the linewidth of the cavity - i.e. $\Omega_m \gg \kappa$ - impressive milestones have been reached regarding the precise control of the mechanical oscillator, including ground-state cooling [15, 16] and squeezing [17–19]. However, these correspond to weak-coupling situations where the intrinsically nonlinear optomechanical coupling may be linearised, and so Gaussian input states are bound to remain Gaussian. Effects of nonlinearity in the resolved sideband regime have been investigated theoretically [20, 21], leading to various proposals regarding the generation of nonclassical mechanical states [22–24]. However the single-photon coupling strengths required for such effects to be observable experimentally have not yet been reached in sideband-resolved experimental platforms. Meanwhile, in the unresolved sideband regime ("bad cavity") where $\Omega_m \ll \kappa$, anharmonic behaviour of the mechanical oscillator due to the intrinsic optomechanical nonlinearity has been predicted and observed experimentally. These anharmonic behaviours include static effects like bistability (theory [25] and experiment for optical [26] and microwave [27] frequency cavities), and also dynamical effects like parametric instability giving rise to self-induced oscillations (theory [28, 29] and experiment [30–32]) or even chaos [33]. One naturally wonders whether in this unresolved sideband regime, where stronger single-photon coupling is achievable (e.g. Ref. [34]), one may harness optomechanical nonlinearity to generate non-Gaussian, and maybe even nonclassical, mechanical states.

In this article, we investigate the behaviour of mechanical oscillator in a laser-driven optomechanical system where the mechanical resonating frequency and the single-photon coupling constant are small compared to the decay rate of the cavity field. Using a semiclassical Fokker-Planck description, we first show that the mechanical states are high-occupation thermal states, thus extending the result already known for the linearised case [35] to include nonlinearity, and in analogy to what is found for driven two-level systems strongly coupled to a mechanical oscillator [36]. The high occupation of the mechanical mode is due to the out-of-equilibrium coupling to the quantum electromagnetic fluctuations of the laser field, present even when the environmental temperature is zero. We go on to investigate the static phenomena described by the Fokker Planck equation. In particular, we predict the ability to transition smoothly from a weakly anharmonic, to quartic, and eventually double-well effective potential, via a careful tuning of the system parameters (notably the input power and detuning of the laser). We investigate the observable signatures of the resulting non-Gaussian mechanical steady states, via measurements of either the mechanics directly or the cavity/outgoing light field. The anharmonic signatures explored in this work include the shifting and broadening of spectral features, as well as a cavity population suppression resembling the suppression of electrical conductance predicted in an electromechanical system in the strong coupling regime [37]. Even though the behaviour we explore is semiclassical, the presence and tunability of strong anharmonicity in mechanical systems represent an important step in the context outlined above.

The remainder of the article is structured as follows. In Sec. 2 we derive a semiclassical model based on a Fokker Planck equation for the motion of the mechanical oscillator. We check the model self-consistently by defining an effective temperature, which is large enough for the parameters considered to justify the semiclassical assumption. In Sec. 3 we discuss conditions for the mechanics to enter regimes of static bistability and dynamical instability. We consider a special parameter tuning to achieve a smooth transition from a harmonic, to a weakly anharmonic, to a purely quartic and eventually to a double-well effective potential. We investigate in Sec. 4 some observable signatures of the mechanical anharmonicity on the behaviour of the oscillator itself. We calculate both analytically and numerically the oscillator displacement spectrum (power spectral density) and also that of the cavity light field. Finally, in Sec. 5 we also reveal some observable imprints of the mechanical anharmonicity left on the cavity light field. In particular, we calculate numerically the cavity population and emission spectrum which both provide information regarding the specific tuning of the mechanical anharmonicity that we consider.

## 2 Model

### 2.1 Input-output equations of motion

The unitary dynamics of a standard cavity OM system is described in the frame rotating at the coherent laser drive frequency by the Hamiltonian [14]:

$$\hat{H}/\hbar = -\Delta \hat{a}^\dagger \hat{a} + \Omega_m \hat{b}^\dagger \hat{b} - g_0 \hat{a}^\dagger \hat{a} \left( \hat{b} + \hat{b}^\dagger \right) + \varepsilon(\hat{a} + \hat{a}^\dagger), \tag{1}$$

where $\hat{a}$ and $\hat{b}$ are annihilation operators for photons and phonons, respectively, and $\hbar$ is the reduced Planck constant. We denote $\Delta = \omega_L - \omega_c$ the detuning between the laser frequency $\omega_L$ and the bare cavity frequency $\omega_c$, $\Omega_m$ the frequency of the mechanical mode, $g_0$ the single-photon OM coupling strength, and $\varepsilon$ is proportional to the input laser drive strength. We note that the sign of the optomechanical interaction term corresponds to a choice of the direction of oscillator displacement $\hat{x} = x_{\mathrm{zpf}}(\hat{b} + \hat{b}^\dagger)$, in terms of the oscillator zero point fluctuations

$x_{\text{zpf}} = \sqrt{\hbar/(2m\Omega_m)}$ with mass $m$. Both signs are used in the literature and this choice has no effect on the predicted physical behaviours.

The Heisenberg equations of motion accounting for noise and dissipation for the cavity and mechanical fields may be derived via input-output theory [14, 38]:

$$\dot{\hat{a}} = \left[ i \left( \Delta + g_0 \hat{x}/x_{\text{zpf}} \right) - \kappa/2 \right] \hat{a} - i\varepsilon + \sqrt{\kappa}\hat{a}_{\text{in}}, \tag{2}$$

$$\dot{\hat{b}} = -\left[ i\Omega_m + \Gamma_m/2 \right] \hat{b} - ig_0 \hat{a}^\dagger \hat{a} + \sqrt{\Gamma_m}\hat{b}_{\text{in}}, \tag{3}$$

where the cavity and mechanical decay rates are $\kappa$ and $\Gamma_m$, respectively, and dot indicates differentiation with respect to time. The input cavity field $\hat{a}_{\text{in}}(t)$ verifies the noise correlations

$$\langle \hat{a}_{\text{in}}(t)\hat{a}_{\text{in}}^\dagger(t') \rangle = (\bar{n}_{\text{th}}^a + 1)\delta(t - t'), \tag{4a}$$

$$\langle \hat{a}_{\text{in}}^\dagger(t)\hat{a}_{\text{in}}(t') \rangle = \bar{n}_{\text{th}}^a \delta(t - t'), \tag{4b}$$

where $\bar{n}_{\text{th}}^a$ is the occupation of the thermal electromagnetic environment. The noise correlators for the input mechanical field $\hat{b}_{\text{in}}(t)$ are identical to those in Eq. (4) but with the corresponding occupation of the thermal mechanical environment $\bar{n}_{\text{th}}^b$. Here, we typically consider $\bar{n}_{\text{th}}^a \to 0$, i.e. the electromagnetic vacuum. This is a standard condition that is not challenging to fulfill for optical cavities, and may require cooling to mK temperatures for microwave resonators. Meanwhile, $\bar{n}_{\text{th}}^b$ will typically be nonzero, since even for MHz mechanical resonators, one would require micro-Kelvin environmental temperatures to reach $\bar{n}_{\text{th}}^b \lesssim 1$. The quantum Langevin equations (2) and (3) are nonlinear operator equations, which may not generally be solved exactly, neither analytically nor numerically.

## 2.2 Semiclassical unresolved sideband regime

We specify to the unresolved sideband regime, where $\kappa \gg \Omega_m$, so the cavity field reacts instantaneously to the mechanics. We assume also that even if $g_0$ can be larger than $\Omega_m$ it remains always much smaller than $\kappa$. In the analogous parameter regime for a mechanical oscillator coupled to a two-level system, it has been shown that the mechanical oscillator mode behaves classically, and acquires a large average occupation [36]. We proceed in a similar way by using a classical description of the mechanical degree-of-freedom, and verify self-consistently that its effective temperature is in the classical range (see Sec. 3.3). From Eqs. (2) and (3) we formulate a semiclassical equation of motion for the mechanical displacement

$$m\ddot{x} = -m\Omega_m^2 x - m\Gamma_m \dot{x} + f + \delta f + \delta f_{\text{th}}, \tag{5}$$

where $f = 2mx_{\text{zpf}}\Omega_m g_0 \langle \hat{a}^\dagger \hat{a} \rangle$ is the average radiation pressure force with fluctuations $\delta f$, and $\delta f_{\text{th}}$ is the thermal noise force due to the thermal mechanical environment ($\hat{b}_{\text{in}}$).

To evaluate the average radiation force $f$, we assume the cavity field consists of a coherent amplitude $\alpha(t)$ and a fluctuating quantum field $\hat{\tilde{a}}(t)$, i.e. $\hat{a}(t) = \alpha(t) + \hat{\tilde{a}}(t)$. From the solutions of the equation of motion (2) [see Apps A.1-A.2 for derivation], we obtain the average radiation pressure force $f(x) = f_0(x) - m\Gamma_{\text{opt}}(x)\dot{x}$, where the static force reads

$$f_0(x) = 2mx_{\text{zpf}}\Omega_m g_0 n(x), \tag{6}$$

and the induced optical damping

$$\Gamma_{\text{opt}}(x) = -4\Omega_m g_0^2 \Delta'(x)\kappa \frac{n(x)}{(\Delta'^2(x) + \kappa^2/4)^2}. \tag{7}$$

Both are given in terms of the position-dependent average number of photons circulating in the cavity

$$n(x) = n_{\text{max}} \frac{\kappa^2/4}{\Delta'^2(x) + \kappa^2/4},\tag{8}$$

where the mechanical displacement enters the effective detuning $\Delta'(x) = \Delta + g_0 x/x_{\text{zpf}}$ and $n_{\text{max}} = 4\varepsilon^2/\kappa^2$ is the resonant cavity photon number characterizing the input laser power $P = \hbar\omega_L n_{\text{max}}\kappa/4$. The expression for the optical damping found here (7) is equivalent to the one found by treating the linearised system quantum mechanically [14]. The Langevin equation (5) may now be explicitly written as

$$m\ddot{x} = -m\Omega_m^2 x - m\Gamma_{\text{tot}}(x)\dot{x} + f_0(x) + \delta f_0 + \delta f_{\text{th}},\tag{9}$$

where $\Gamma_{\text{tot}} = \Gamma_m + \Gamma_{\text{opt}}$ and the optical noise term $\delta f_0$ characterises the photonic noise imprinted on the oscillator motion due to the optomechanical interaction. Thus, the mechanical oscillator experiences the static force $F(x) = -m\Omega_m^2 x + f_0(x)$ or, equivalently, an effective potential $V_{\text{eff}}$ defined via $F = -\partial V_{\text{eff}}/\partial x$. Writing the effective potential explicitly we find

$$\frac{2V_{\text{eff}}(x)}{\hbar\kappa} = \frac{1}{\lambda}\left(\frac{\Delta'(x) - \Delta}{\kappa}\right)^2 - n_{\text{max}}\arctan\left[\frac{2\Delta'(x)}{\kappa}\right],\tag{10}$$

where we introduced the single-photon parametric cooperativity $\lambda = 2g_0^2/(\Omega_m\kappa)$ [34]. This parameter corresponds to what in the condensed matter field is usually called the polaronic energy $\hbar g_0^2/\Omega_m$ divided by the cavity damping rate $\kappa$.

## 3 Tuning the anharmonicity

### 3.1 Static bistability and dynamical instability

The equilibrium position(s) $x_e$ of the mechanical oscillator is (are) obtained from the condition $\partial V_{\text{eff}}/\partial x = 0$, when the radiation pressure force balances the intrinsic restoring force. This implies

$$\frac{1}{\lambda\kappa n_{\text{max}}}\left(\Delta'(x_e) - \Delta\right) = \frac{1}{1 + \left(\frac{2\Delta'(x_e)}{\kappa}\right)^2}.\tag{11}$$

This cubic equation (11) for $\Delta'$ has one or three real solutions for a given set of system parameters. In the spirit of Ref. [36], we first differentiate both sides of Eq. (11) with respect to $\Delta'$. The solutions for different values of the input power characterised by $n_{\text{max}}$ are shown by the black line in Fig. 1(a). We then use Eq. (11) to find the region of static bistability as a function of the detuning $\Delta$, as shown in Fig. 1(b). The hatched magenta region denotes where the effective potential $V_{\text{eff}}$ forms a double-well, while outside this region the potential has only one minimum.

In addition to static bistability, the mechanical oscillitor may enter a regime of dynamical instability. The overall mechanical damping rate $\Gamma_{\text{tot}}$ appearing in Eq. (9) may become negative, giving rise to the possibility of motional amplification – i.e. self-sustained oscillations. In Fig. 1(a), we show shaded gray the regions of dynamical instability found by the solutions of the equation $\Gamma_{\text{tot}} = 0$, for different mechanical oscillator finesse, quantified by the quality factor $Q \equiv \Omega_m/\Gamma_m$. The separatrix at $\Delta' = 0$ (gray dotted line) indicates the boundary between the so-called regimes of optomechanical heating ($\Delta' > 0$) and cooling ($\Delta' < 0$). Fig. 1(b) indicates the corresponding regions in terms of the externally tunable parameter $\Delta$, computed using Eq. (11). Note that the minimum value of laser power for which dynamical instability

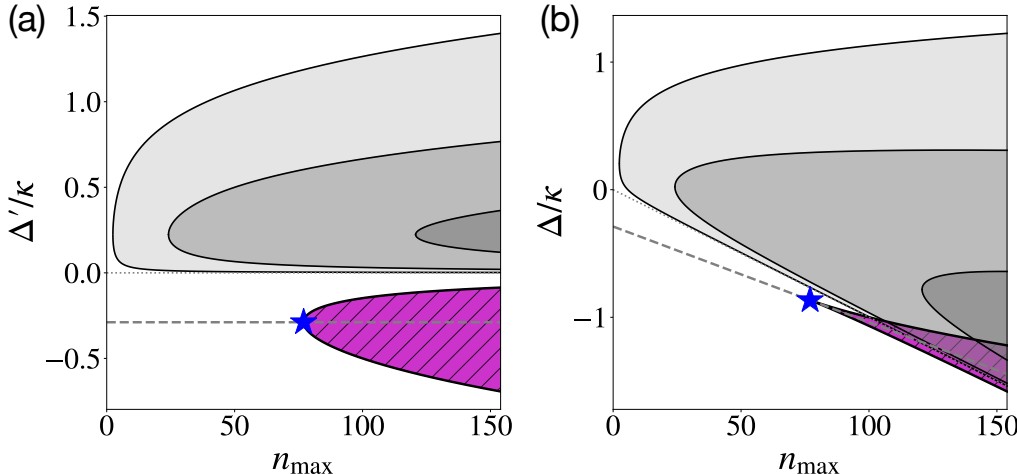

Figure 1: Optomechanical system behaviour as a function of parameters $n_{\text{max}}$ and (a) $\Delta'$ and (b) $\Delta$, indicating the region where the effective potential $V_{\text{eff}}$ forms a double well (hatched magenta), as well as the region of dynamical instability for different values of intrinsic damping (from light to dark gray for $Q \equiv \Omega_m/\Gamma_m = 1000, 100, 20$). In (a) the dotted gray line at $\Delta' = 0$ indicates the separatrix between stable and possibly dynamically unstable regions for $Q \to \infty$, and the dashed gray line indicates the special tuning of the anharmonicity described by $\Delta'_* \equiv -\kappa/(2\sqrt{3})$. The corresponding regions and lines in (b) are found via Eq. (11), where the gray dashed line is described by Eq. (12). The blue star indicates the point in parameter space where the $V_{\text{eff}}$ is purely quartic. In the plot we took $\lambda = 0.01$ and $\kappa/\Omega_m = 100$.

may set in (for any $\Delta$) is set by $n_{\text{max}} = (27\sqrt{5}/500)(\kappa/g_0)^2 Q^{-1}$, so the picture in Fig. 1(b) given for specific $Q$ values may be significantly affected by changing $\kappa$ and $g_0$.

We focus on the smooth transition whereby the potential goes from a single to a double-well, where the new unstable equilibrium position between the two wells remains at the position in $\Delta'$ of the original stable equilibrium position. We wrote Eq. (11) in such a way to put the externally tunable parameters, $\Delta$ and $n_{\text{max}}$, on the left-hand-side. The desired transition may then be achieved by adjusting these parameters such that the central solution to Eq. (11) remains at the steepest point of the Lorentzian on the right-hand-side, which occurs at $\Delta' = \Delta'_* \equiv -\kappa/(2\sqrt{3})$. Ensuring this is an equilibrium position amounts to plugging in to Eq. (11), leading to the linear relation:

$$\Delta = \Delta'_* - (3/4)\kappa\lambda n_{\text{max}}. \tag{12}$$

The definition of $\Delta'_*$ and Eq. (12) define the dashed gray lines in Figs. 1(a) and 1(b), respectively. The effect of this special tuning of the anharmonicity on the solutions of Eq. (11) is depicted graphically in Fig. 2(a). Increasing the laser power $n_{\text{max}}$ while changing the detuning $\Delta$ so as to follow the dashed line in Fig. 1(b) [Eq. (12)] gradually increases the anharmonicity of $V_{\text{eff}}$, as shown in Fig. 2(b).

In fact, the anharmonicity increases along Eq. (12) in a special way. To see this, we write its expansion up to fourth order for small displacements around the position $\Delta'_*$:

$$V_{\text{eff}}/\hbar = V_{\text{eff}}(\Delta'_*) + B(\Delta' - \Delta'_*)^2 + C(\Delta' - \Delta'_*)^4,$$
$$B = \frac{1}{2\kappa}\left(\frac{1}{\lambda} - \frac{3\sqrt{3}}{4}n_{\text{max}}\right), \quad C = \frac{9\sqrt{3}}{16}\frac{n_{\text{max}}}{\kappa^3}. \tag{13}$$

There is never any cubic term in the expansion due to the condition (12). In addition, we see from Eq. (13) that the quadratic (harmonic) term may be totally suppressed by choosing

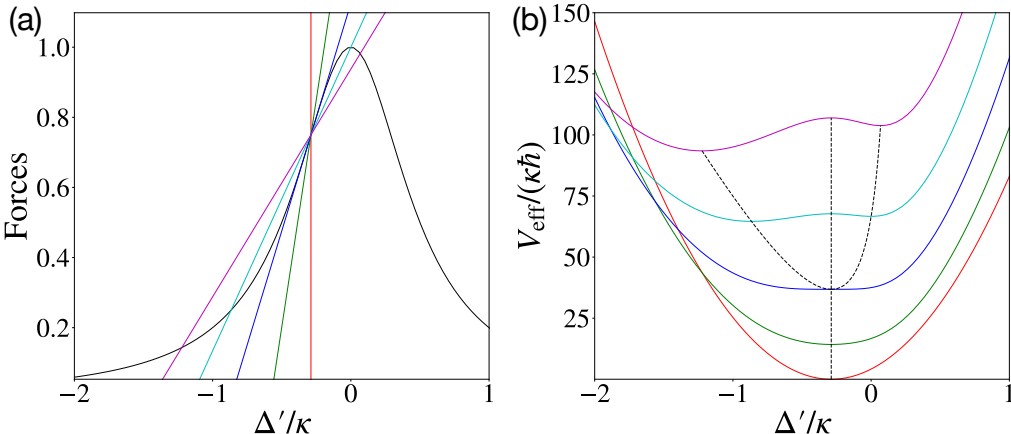

Figure 2: Tuning $n_{\max}$ and $\Delta$ along the line given in Eq. (12) – (a) the two sides of Eq. (11) where the crossings denote oscillator equilibrium positions, and (b) the evolution of the corresponding effective potential (10). The colors from red to magenta are for $n_{\max}/n^*_{\max} = 0, 0.5, 1, 1.5, 2$. In the plot we took $\lambda = 0.01$ giving $n^*_{\max} \approx 77$.

$n_{\max} = n^*_{\max} \equiv 4/(3\sqrt{3}\lambda)$, such that $B = 0$. Here, the quadratic part of the radiation pressure term exactly cancels the harmonic restoring force of the bare mechanical oscillator. This point is marked by the blue star in Figs. 1(a) and 1(b), and the corresponding blue curve in Fig. 2(b), where the potential is approximately quartic around the point $\Delta'_*$. The physical parameters corresponding to this quartic point are therefore $n_{\max} = n^*_{\max}$ and $\Delta = \Delta^* \equiv -\sqrt{3}\kappa/2$. We naturally wonder what are the observable consequences of this special anharmonicity on the system behaviour. We explore this question in the remainder of the paper.

We note here that the conditions for the appearance of the purely quartic potential ($n^*_{\max}$ and $\Delta^*$) coincide with the conditions on $n_{\max}$ and $\Delta$ for the maximum cooling expected in an optomechanical system in presence of an intrinsically nonlinear optical cavity in the same regime of $\kappa \gg \omega_m$ – cf. Eqs. (6) and (7) of Ref. [39]. In that case the bistability is generated by tuning the large intrinsic nonlinearity.

## 3.2 Fokker Planck equation

From the Langevin equation (9) we can derive a Fokker Planck equation for the probability density $P(x, p, t)$ of the oscillator [40]:

$$\partial_t P = \left[ -\frac{p}{m} \partial_x - \left( f_0 - m\Omega_m^2 x \right) \partial_p + \Gamma_{\text{tot}} \partial_p p + \frac{D_{\text{tot}}}{2} \partial_p^2 \right] P, \tag{14}$$

where we introduced the momentum $p = m\dot{x}$. The total diffusion coefficient reads $D_{\text{tot}} = D_{\text{th}} + D_{\text{opt}}$, where $D_{\text{th}}$ ($D_{\text{opt}}$) is the spectral noise weight of $\delta f_{\text{th}}$ ($\delta f_0$). For the thermal Brownian noise we have $D_{\text{th}} = m\Gamma_m(\hbar\Omega_m)\coth(\hbar\Omega_m/2k_B T_b)$ where $T_b$ is the temperature of the mechanical environment. For the noise inherited by the mechanical oscillator from the optics we find [see App. A.3]:

$$D_{\text{opt}} \approx 2m\hbar\Omega_m g_0^2 \kappa \frac{n(x)}{\Delta'^2 + (\kappa/2)^2}, \tag{15}$$

where we assumed $\Omega_m \ll \Delta'$. As shown in App. A.3, the fluctuation term (15) is due to the quantum electromagnetic fluctuations of the laser field. The Fokker Planck equation (14) describes the semiclassical behaviour of a mechanical oscillator in an optomechanical system. In the remainder of the paper we extract certain aspects of the physics contained in Eq. (14).

### 3.3 Oscillator effective temperature

From the Fokker-Plank equation (14), we can assign the effective temperature of the mechanical oscillator and in doing so check the assumption of treating the oscillator displacement as a classical variable in the unresolved sideband regime, where $\kappa \gg \Omega_m$. By the fluctuation-dissipation theorem, we define the effective temperature of the mechanical oscillator $T_{\text{eff}}$ according to

$$\coth\left(\frac{\hbar\Omega_m}{2k_B T_{\text{eff}}}\right) = \frac{D_{\text{tot}}}{m\Gamma_{\text{tot}}\hbar\Omega_m}. \tag{16}$$

Presuming that the optical parts dominate both the fluctuations and the dissipation, we find

$$\coth\left(\frac{\hbar\Omega_m}{2k_B T_{\text{eff}}}\right) = -\frac{\Delta'^2 + \kappa^2/4}{2\Omega_m\Delta'}. \tag{17}$$

We focus on the parameter range where $\Delta' = \Delta'_* \sim \kappa \gg \Omega_m$, so the right-hand-side is large implying $k_B T_{\text{eff}} \gg \hbar\Omega_m$. The average number of phonons in the oscillator $n_b = (e^{\hbar\Omega_m/k_B T_{\text{eff}}} - 1)^{-1} \gg 1$ is therefore large, corresponding to a state far from the quantum ground state. This justifies self-consistently the validity of the semiclassical description given. A possible interpretation of this effect is that the poissonian quantum fluctuations of the number of photons in the cavity strongly coupled to the external environment ($\kappa \gg g_0, \Omega_m$) generate a large effective fluctuating force acting on the mechanical oscillator leading to large displacement fluctuations, or equivalently, a large effective temperature.

## 4 Signatures of anharmonicity on mechanics

### 4.1 Stationary probability distribution

The Fokker Planck equation (14) may be solved for its stationary solution. Writing Eq. (14), as $\partial_t P = \mathcal{L}P$, we have that $\mathcal{L}P^{\text{st}} = 0$, with $P^{\text{st}}$ also satisfying a normalization condition. We solve for $P^{\text{st}}$ numerically [36,41] by discretizing the $x-p$ phase space on a grid of dimensions $N_x \times N_p$; the operator $\mathcal{L}$ therefore becomes a matrix of size $N = N_x N_p$. Once a reasonable window of $x-p$ phase space is defined, values of $N_x = N_p = 100$ are generally sufficient to obtain the solution for $P^{\text{st}}$.

We find that for the parameters considered the stationary state of this driven-dissipative (i.e. nonequilibrium) system is nonetheless described by a Gibbs distribution. In other words, we may write $P^{\text{st}}(x, p) = \mathcal{N}\exp[-E(x, p)/k_B T_{\text{eff}}]$, where $E(x, p) = p^2/2m + V_{\text{eff}}(x)$, $T_{\text{eff}}$ is determined by Eq. (16), and $\mathcal{N}$ ensures normalization: $\int dx \int dp\, P^{\text{st}}(x, p) = 1$. An example of the stationary state distributions are shown in Fig. 3(a-c), for the case where the potential is purely quartic ($n_{\text{max}} = n^*_{\text{max}}$), indicating good agreement between numerical and analytical results. In particular, the flat bottom of the quartic potential is manifested in the marginal distribution $P^{\text{st}}(x)$ shown in Fig. 3(a). The slight asymmetry present in the numerical result for $P^{\text{st}}(x)$ is likely due to 5th (and higher) order terms that are neglected in our potential expansion (13).

### 4.2 Mechanical displacement spectrum $S_{zz}(\Omega)$

We define the auto-correlation function for general position-dependent, zero-average operators $A(x)$ and $B(x)$ as $S_{AB}(t > 0) = \langle A[x(t)]B[x(0)]\rangle$. The angle brackets denote the average over the mechanical system's stationary state, that is described by the probability distribution $P^{\text{st}}$. The spectrum, defined as the Fourier transform of the correlation function,

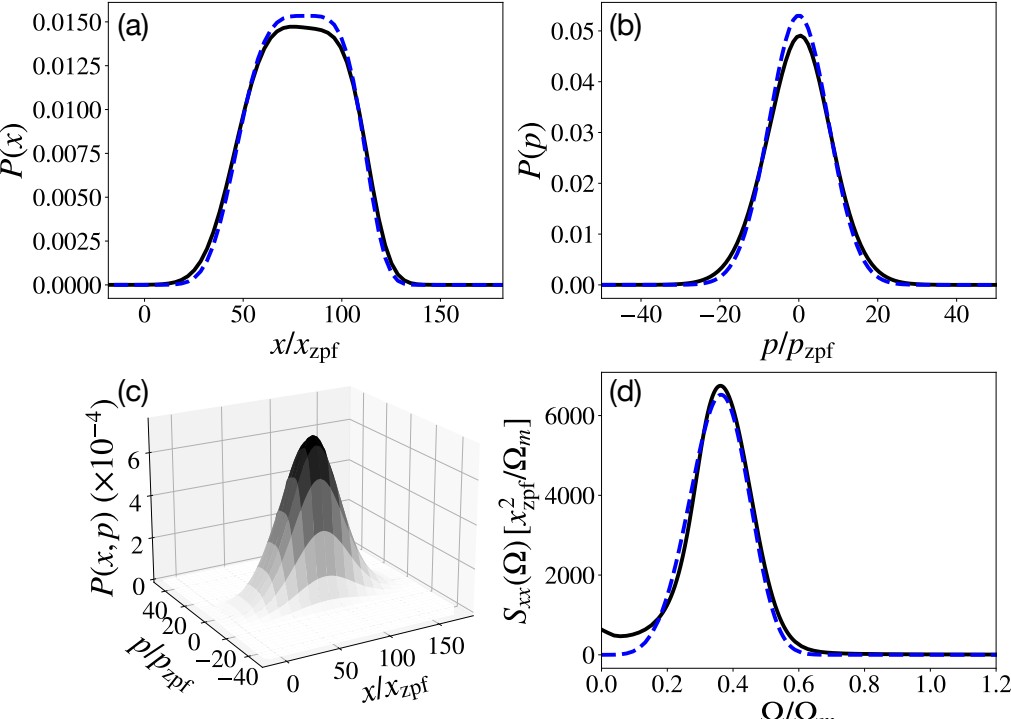

Figure 3: The stationary probability distribution at the point $n_{\max} = n^*_{\max}$. We show the full 2d distribution $P^{\mathrm{st}}(x, p)$ (c) and its marginals $P^{\mathrm{st}}(x)$ (a) and $P^{\mathrm{st}}(p)$ (b), found by integrating over $p$ and $x$, respectively. The solid lines are results of numerical simulation of the the Fokker-Planck equation (14). The dashed lines are theoretical results taking for $T_{\mathrm{eff}}$ the expression in Eq. (16). Panel (d) shows the displacement spectrum, Eq. (20). The parameters are $\lambda = 0.01$, $\kappa/\Omega_m = 100$, $\Gamma_m/\Omega_m = 0.001$, and $k_B T_b/\hbar\Omega_m = 10$.

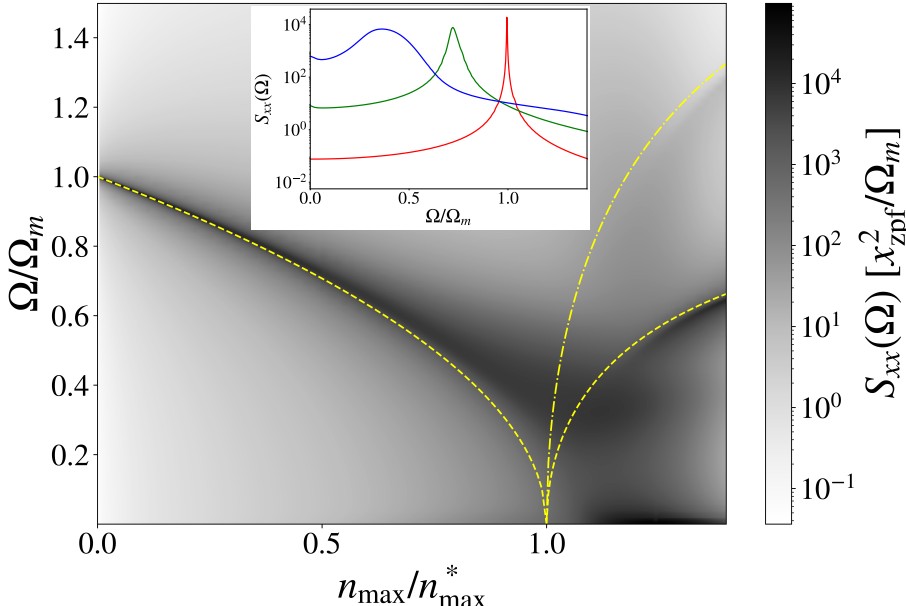

Figure 4: The displacement spectrum as the system anharmonicity is swept through the quartic transition – i.e. as $n_{\max}$ is increased $\Delta$ is changed in a commensurate way such that Eq. (12) is always satisfied. The yellow lines correspond to the predicted mechanical frequency renormalization, defined in the text by $\bar{\Omega}_m$ (dashed), and the second harmonic $2\bar{\Omega}_m$ (dot-dashed). Inset shows cuts at $n_{\max}/n^*_{\max} = 0.005$ (red), 0.5 (green) and 1 (blue). The other parameters are as in Fig. 3.

$S_{AB}(\Omega) = \int_{-\infty}^{\infty} dt\, e^{i\Omega t} S_{AB}(t)$, may be expressed by the discretized formula

$$S_{\sigma\sigma}(\Omega) = -2 \sum_{ij} A(x_i) \left[ \frac{\mathcal{L}}{\Omega^2 + \mathcal{L}^2} \right]_{ij} B(x_j) P_j^{\mathrm{st}}, \tag{18}$$

where each index $i$ labels a state with a certain discrete set of stochastic variables $x_i$ and $p_i$. Quantities of the type Eq. (18) may be straightforwardly calculated numerically [41].

We now turn to the mechanical displacement spectrum, $S_{xx}(\Omega)$. Taking $A(x) = B(x) = \tilde{x}(t) \equiv x(t) - \langle x \rangle$ in Eq. (18), we compute numerically $S_{xx}(\Omega)$ for many values of the input laser power, so as to sweep through the quartic transition – i.e. we change $n_{\max}$ and $\Delta$ according to Eq. (12) following the gray line in Fig. 1(b). The resulting spectra are shown in the heat map in Fig 4, from which we extract some notable features. Firstly the renormalization of the mechanical resonance frequency with increasing laser power may be understood by the optical spring effect. Indeed, taking the total force $F(x) = -m\Omega_m^2 x + f_0(x)$, we may define the (harmonic) renormalized frequency $\bar{\Omega}_m$ via the expression $m\bar{\Omega}_m^2 = -(dF/dx)_{x=x_{\mathrm{eq}}}$, where $x_{\mathrm{eq}}$ is a stable equilibrium position satisfying $F(x_{\mathrm{eq}}) = 0$. We find that the expression reads $\bar{\Omega}_m/\Omega_m = \sqrt{1 - \tilde{n}_{\max}}$ for $\tilde{n}_{\max} \equiv n_{\max}/n^*_{\max} < 1$ and $\sqrt{2(\tilde{n}_{\max} - 1)}$ for $\tilde{n}_{\max} \gtrsim 1$. As shown by the yellow curves in Fig. 4 these expressions capture rather well the spring-softening due to the increasing radiation pressure force. After the transition to the double-well regime, for $n_{\max} > n^*_{\max}$, the resonance frequency starts again to increase as the separation and barrier height between the wells increase [see Fig. 2(b)]. The additional low-frequency contribution to $S_{xx}(\Omega)$ present at $n_{\max} > n^*_{\max}$ is due to the slow switching dynamics between the two stable equilibria. The second harmonic $2\bar{\Omega}_m$, indicated in Fig. 4 by the dot-dashed yellow line, is also visible in the bistable region.

As well as frequency renormalization, the spectrum also undergoes drastic changes in its line shape, as highlighted by the cuts shown in the inset of Fig. 4. In order to understand the

broadening of the mechanical spectrum, we may calculate it analytically. The calculation of the displacement spectrum for a Duffing oscillator was first performed in Ref. [42]. Recently, this calculation was applied to an electromechanical system of a mechanical oscillator coupled to a quantum dot [43]. We apply it here to our optomechanical system, described by the (Duffing) anharmonic potential Eq. (13).

Following Refs [42, 43], we neglect the damping and diffusion terms in Eq. (14). In doing so we hypothesise that the oscillator performs many oscillations on the closed phase space trajectory with constant energy $E = E(x(t), p(t))$ before drifting to another trajectory. The slow trajectory transition rate is given by the average dissipation coefficient, $\gamma = \int dx \int dp\, P^{\text{st}}(x,p)\Gamma_{\text{tot}}(x)$. The fixed-energy oscillation frequency may be calculated by integrating over a period, giving $\omega(E) = 1/\left[2\pi\sqrt{m/2}\oint (E - V_{\text{eff}}(x))^{-1/2}\, dx\right]$, where the anharmonicity in $V_{\text{eff}}(x)$ induces the dispersion in $\omega(E)$, and the previously introduced $\bar{\Omega}_m$ corresponds to $\omega(0)$. Thermal fluctuations will populate all excitation trajectories up to those corresponding to $E = k_B T_{\text{eff}}$, leading to an overall frequency spread $\Delta_\omega \sim \omega(k_B T_{\text{eff}}) - \bar{\Omega}_m$. If $\Delta_\omega \gg \gamma$ we expect the resulting spectral features will be dominated by the anharmonicity, and not the damping, and vice versa in the opposite limit [44].

The displacement spectrum for the dissipationless anharmonic oscillator may be written as (see App. B.1 for derivation):

$$S_{xx}(\Omega) = \mathcal{N}(2\pi)^2 \sum_{n=0}^{\infty} \frac{e^{-E_n/k_B T_{\text{eff}}}}{n\omega(E_n)\,|\partial\omega(E)/\partial E|_{E_n}}\tilde{x}_n^2(E_n), \tag{19}$$

where $\tilde{x}_n(E)$ are the Fourier series coefficients of the (zero-average) fixed-energy mode trajectories given by $\tilde{x}_E(t) = \sum_n e^{in\omega(E)t}\tilde{x}_n(E)$, and $E_n$ are the discrete energy levels satisfying the equation $n\omega(E_n) = \Omega$. The problem of calculating the spectrum is therefore reduced to finding the Fourier series components $x_n(E)$ and the mode frequency $\omega(E)$. In App. B we perform the calculation for a general anharmonic potential of the form $V(x) = \mu x^2 + \nu x^4$, for $\mu$ and $\nu$ constants. One relevant limiting case is where the quadratic term vanishes ($\mu = 0$) leaving a purely quartic potential. We focus on this since it is what we expect for our optomechanical system at $n_{\text{max}} = n_{\text{max}}^*$ and $\Delta = \Delta^*$, as discussed in Sec. 3. Plugging in the expression for $\nu$ from Eq. (13) [see App. B.3] we find that the normalised spectrum may be written in the compact form

$$\frac{S_{xx}(\Omega)}{S_{xx}(\Omega_{\text{max}})} = \left(\frac{\Omega}{\Omega_{\text{max}}}\right)^4 e^{-(\Omega/\Omega_{\text{max}})^4 + 1}, \tag{20}$$

which is peaked at $\Omega_{\text{max}} = (\pi/2K[-1])[3\lambda k_B T_{\text{eff}}/\hbar\kappa]^{1/4}\Omega_m$, where $K[-1] \approx 1.31$ is the complete elliptic integral of the first kind. The full width half maximum of the spectrum (20) is given by $\delta\Omega \approx 0.585\Omega_{\text{max}}$, leading to the effective quality factor $\Omega_{\text{max}}/\delta\Omega = 1.71$, independent of any other system parameters. The dissipationless analytical expression (20) is shown alongside the full numerical solution in Fig. 3(d), showing good agreement for the line shape, peak position and width, as well as the overall noise magnitude. For the parameters chosen the spectral width due to the anharmonicity $\delta\Omega$ is indeed large compared to $\gamma$, hence explaining the accuracy of Eq. (20).

The expression (20) relies on the fact that at the critical point $(n_{\text{max}}^*, \Delta^*)$ the truncated expansion of $V_{\text{eff}}$ at fourth order (13) remains valid. We can use this fact to develop an additional condition of validity for the expression. Namely, we require that the average displacement fluctuations of the oscillator are small compared to the width of the quartic potential well. In other words, we require $(g_0/\kappa)\langle\delta x^2\rangle/x_{\text{zpf}}^2 \ll 1$, where $\langle\delta x^2\rangle = S_{xx}(t=0)$ is the variance. Plugging in the expression for the quartic potential to calculate the variance leads to the condition

$$\sqrt{\frac{1}{2\sqrt{3}}\frac{\Gamma[3/4]}{\Gamma[5/4]}}\left(\lambda\frac{k_B T_{\text{eff}}}{\hbar\kappa}\right)^{1/4} \ll 1, \tag{21}$$

where $\Gamma[x]$ is the complete Gamma function. From Eq. (17) we have $k_B T_{\text{eff}}/\hbar \approx \kappa/(2\sqrt{3})$. This indicates that the anharmonic (quartic) signature is more visible for weaker optomechanical coupling, as corroborated by Fig. 3(d) which is for $\lambda = 0.01$.

We briefly mention the opposite limit of the anharmonic potential $V(x) = \mu x^2 + \nu x^4$, where the quartic term remains small compared to the quadratic one ($\langle \nu x^4 \rangle \ll \langle \mu x^2 \rangle$). Here, it is also possible to extract a simple analytical expression for the spectrum (see App. B.2), which has a non-Lorentzian asymmetric line shape. However, the numerical results indicate a Lorentzian line shape where one may expect this weakly anharmonic result to apply – see red and green curves in inset of Fig. 4. We verify that for the parameters considered the predicted spectral width due to the anharmonicity is small compared to the average dissipation rate [see expression (B.11)], and so it is unsurprising that the spectrum is dominated by the damping [14].

Although some optomechanics experimental setups give access allow direct measurement of the mechanical displacement [34], most measure the light field exiting the cavity. In these cases, where a measurement of $S_{xx}(\Omega)$ may be challenging, one may nonetheless hope to observe signatures of the mechanical anharmonicity in the light field.

## 5 Imprints of anharmonic mechanics on light field

### 5.1 Cavity population $S(\Delta)$

By input-output theory it may be shown that the output light field picked up by a detector in an experiment will reflect exactly the statistics of the cavity light field [45]. We therefore explore here the properties of the cavity light field itself, for simplicity. The average population of the cavity depends on the laser detuning, and may be quantified by the (normalised) intensity $S(\Delta) = \lim_{t \to \infty} \langle \hat{a}^\dagger(t) \hat{a}(t) \rangle / n_{\text{max}}$, where $n_{\text{max}}$ is (as previously defined) the number of photons present for a laser drive on resonance with the cavity frequency ($\Delta = 0$). Due to the separation of timescales in the parameter-regime-of-interest $\kappa \gg \Omega_m$, we may approximate the cavity field by its static solution in Eq. (2). The slow dynamics of $x(t)$ are nevertheless taken into account, via the solution to the Fokker Planck equation $P^{\text{st}}(x, p)$. In other words, we take the approximate solution for the cavity field:

$$\alpha[x(t)] \approx \frac{i\varepsilon}{i(\Delta + g_0 x(t)/x_{\text{zpf}}) - \kappa/2}, \tag{22}$$

where $x(t)$ may be considered an external variable, that behaves according to the Fokker-Planck equation (14). We have neglected the term proportional to $\hat{a}_{\text{in}}$ since we always consider a low occupation of the electromagnetic environment $\bar{n}_{\text{th}}^a \ll 1$, and consequently rewritten $\hat{a} \to \alpha$ to stress that it is no longer an operator. The population may then be approximated as:

$$S(\Delta) = \int dx \int dp \, P^{\text{st}}(x, p) \frac{(\kappa/2)^2}{(\Delta + g_0 x/x_{\text{zpf}})^2 + (\kappa/2)^2}. \tag{23}$$

Equation (23) indicates how the oscillator anharmonicity may affect - via the $x$-dependence of $P^{\text{st}}$ and the denominator - the overall behaviour of the optical cavity.

Fixing $n_{\text{max}} = n_{\text{max}}^*$, we sweep $\Delta$ and calculate $P^{\text{st}}$ numerically for each value. The resulting $S(\Delta)$ are shown in Fig. 5 for different values of mechanical quality factor $Q \equiv \Omega_m/\Gamma_m$. We note some features of the cavity population curves shown in Fig. 5. For all values of $\Gamma_m$ we see the expected bending of the cavity line shape from its bare Lorentzian form, reflecting the cavity's effective anharmonicity due to the optomechanical interaction. In all cases the gradient of $S(\Delta)$ becomes infinite exactly at $\Delta = \Delta^*$, indicating being on the cusp of entering

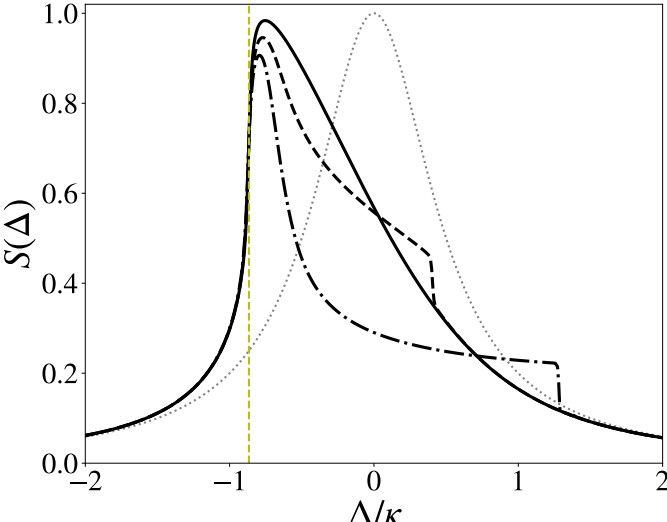

Figure 5: Cavity population calculated numerically via Eq. (23). All black curves share the parameters $n_{\max} = n_{\max}^*$, $\lambda = 0.01$, $\kappa/\Omega_m = 100$, and $k_B T_b/\hbar\Omega_m = 10$ (as in Fig. 3), and we take for the mechanical oscillator finesse $Q \equiv \Omega_m/\Gamma_m = 1000$ (dot-dashed), 100 (dashed) and 20 (solid). The dotted gray line is for a bare cavity ($\lambda = 0$), while the vertical dashed yellow line indicates the critical detuning $\Delta = \Delta^* \equiv -\sqrt{3}\kappa/2$

the bistable regime [see blue star in Fig. 1(b)], and hence providing a signature of being at $n_{\max} = n_{\max}^*$ where $V_{\text{eff}}$ becomes purely quartic. For $n_{\max} > n_{\max}^*$ the bend in $S(\Delta)$ will increase further resulting in a double-valued form for $S(\Delta)$, as given by the hatched magenta region in Fig. 1(b).

The difference in the three black curves is due to the varying prominence of the dynamical instability, as a function of oscillator intrinsic dissipation (as predicted already for the same $Q$ values in Fig. 1(b), indicated by the shaded gray regions). For the larger $Q$ values (dot-dashed and dashed curves), the mechanical oscillator enters the dynamically unstable regime as $\Delta$ is swept above $\Delta^*$. The coincidence of all the curves at the highest values of $\Delta$ indicates the return to the stable regime, corroborating the prediction in Fig. 1(b). That this return comes about via steep jumps in the line shapes suggests another type of bistability between two states – i.e. a self-sustained oscillation state and a static (fluctuating) state. A measured signal for $S(\Delta)$ may therefore present a hysteresis around this transition according to whether $\Delta$ is scanned up or down.

## 5.2 Cavity emission spectrum $S_{a^\dagger a}(\Omega)$

Another related and relevant observable for the cavity light field is the frequency-resolved emission spectrum, whose definition is given by

$$S_{a^\dagger a}(\Omega) = \frac{1}{2\pi n_{\max}} \int_{-\infty}^{\infty} dt\, e^{i\Omega t} \langle \hat{a}^\dagger(t)\hat{a}(0)\rangle. \tag{24}$$

One can easily notice the connection between $S_{a^\dagger a}(\Omega)$ and the previously discussed cavity population $S(\Delta)$. Namely, we have that $\int_{-\infty}^{\infty} S_{a^\dagger a}(\Omega)\,d\Omega = S(\Delta)$. In words, $S(\Delta)$ is a measure of the total number of cavity photons, regardless of their frequency, while $S_{a^\dagger a}(\Omega)$ provides a detailed breakdown based on the energy of the photons. While all photons arrive with the

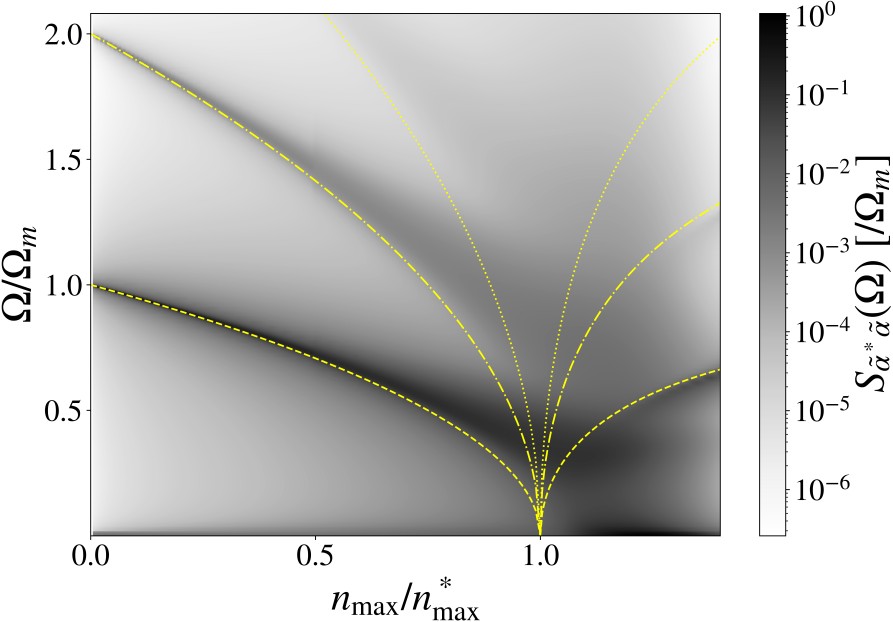

Figure 6: Cavity emission spectrum $S_{\tilde{\alpha}^*\tilde{\alpha}}(\Omega)$ calculated numerically via Eq. (18). The parameters are as for the oscillator displacement spectrum in Fig. 4 (see caption Fig. 3). The yellow lines correspond to the renormalized mechanical frequency, defined in the text in Sec. 4.2 by $\bar{\Omega}_m$ (dashed), and the higher harmonics $2\bar{\Omega}_m$ (dot-dashed) and $3\bar{\Omega}_m$ (dotted).

same energy set by the drive frequency, this energy may be modified due to interaction with the mechanical oscillator, resulting in a spectral distribution described by $S_{a^\dagger a}(\Omega)$.

Once again exploiting the fact that the cavity reacts very fast to the oscillator motion ($\kappa \gg \Omega_m$), we approximate the cavity field by Eq. (22). We may further decompose this coherent part of the cavity field into an average and a fluctuating (zero-average) part, i.e. $\alpha(x) = \bar{\alpha} + \tilde{\alpha}(x)$, where $\bar{\alpha} = \int dx \int dp\, P^{\text{st}}(x,p)\alpha(x)$. The full emission spectrum (24) therefore contains two contributions: $S_{a^\dagger a}(\Omega) = \delta(\Omega)|\bar{\alpha}|^2/n_{\max}+S_{\tilde{\alpha}^*\tilde{\alpha}}(\Omega)$, where $\delta$ is the Dirac delta function and the latter term is defined according to Eq. (24) but with $\hat{a}$ replaced by $\tilde{\alpha}$. Before discussing the $\delta$-correlated peak, we first apply the general formula (18) to compute numerically the spectrum of the fluctuating part, $S_{\tilde{\alpha}^*\tilde{\alpha}}(\Omega)$, according to $P^{\text{st}}$. The resulting spectrum is shown in Fig. 6 where we tune the laser power and detuning in order to satisfy Eq. (12) [gray dashed line in Fig. 1(b)], just as for the displacement spectrum $S_{xx}(\Omega)$ shown in Fig. 4. The cavity emission spectrum $S_{\tilde{\alpha}^*\alpha}(\Omega)$ (Fig. 6) contains many of the same features as the displacement spectrum $S_{xx}(\Omega)$ (Fig. 4), reflecting the fact that the cavity field reacts quickly to follow the dynamics of the mechanical oscillator ($\kappa \gg \Omega_m$). The similar renormalization and broadening of the principle spectral peak indicates that the mechanical anharmonicity discussed in previous sections leaves a clear signature in the light field. The harmonic features at $\Omega = 2\bar{\Omega}_m$ (dot-dashed yellow), $\Omega = 3\bar{\Omega}_m$ (dotted yellow) and also $\Omega = 0$ are visible both before and after the bistable transition, due to the nonlinear dependency on $x$ of Eq. (22).

As previously mentioned, the full cavity output spectrum (24) contains an additional $\delta$-correlated peak at $\Omega = 0$, which corresponds to the laser drive frequency $\omega_L$ in the laboratory frame. This coherent (or elastic) contribution describes the photons that enter the cavity and do not have their energy modified by the interaction with the mechanical oscillator. We verify that the sum of this coherent contribution with the incoherent one coming from $S_{\tilde{\alpha}^*\tilde{\alpha}}(\Omega)$ does indeed return the total cavity population, $S(\Delta)$, as shown by the good match between the solid black and dashed magenta curves in Fig. 7. Since we once again follow the dashed

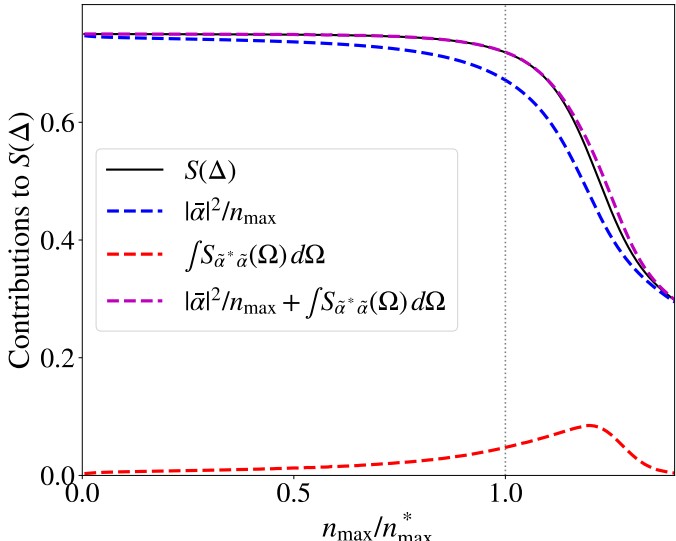

Figure 7: Coherent (dashed blue) and incoherent (dashed red) contributions to the cavity population. Their sum (dashed magenta) agrees with the value obtained for $S(\Delta)$ directly from the expression (23) (solid black). The detuning $\Delta$ is modulated as $n_{max}$ is increased such that Eq. (12) is satisfied [dashed gray line in Fig. 1(b)], as in Figs. 4 and 6. Other parameters are also as in those figures (see caption Fig. 3).

gray line of Fig. 1(b) as $n_{max}$ is increased, $V_{eff}$ forms two minima for $n_{max} > n_{max}^*$ [we enter into the magenta region of Fig. 1(b)]. Here we observe a strong reduction in the overall population of the cavity (solid black line in Fig. 7). This reduction bears strong similarity with the reduction of the conductance predicted for a single-electron transistor coupled to a mechanical oscillator in the strong electro-mechanical coupling limit and in the slow oscillator regime, when the oscillator frequency is much smaller than the tunneling rate [37, 41] (cf. for instance Fig. 3 of Ref. [37]). This behaviour has recently been observed in suspended carbon nanotubes [46]. For a coupling larger than the critical one the system potential develops two symmetric minima for which the resonant condition is not perfectly satisfied, since the oscillator displaces with respect to the position for which the cavity is in resonance. The stronger the parametric coupling, or the drive, the deeper the minima and the stronger the reduction of the number of photons in the cavity. In the opposite fast oscillator regime the effect has been termed as Franck-Condon blockade [47], due to the presence of characteristic steps in the current signal, and has been observed also in carbon nanotubes [48]. In this work we predict a similar population suppression even for weak coupling ($\lambda = 0.01$ in Fig. 7) in a cavity optomechanical system. Note that due to the semiclassical behaviour of the mechanical oscillator photon blockade is not possible, i.e. the quantum nonlinearity is not effective in preventing excitation of many-photon states in the cavity, as instead predicted for the resolved sideband regime [20]. On the other side a very counter-intuitive effect of reduction of occupation of the cavity when increasing the driving is expected. The transition is also indicated by an initial increase and subsequent maximum in the finite-frequency (incoherent) fluctuations (red dashed line in Fig. 7), which is a typical signature of a transition from monostable to bistable behaviour. Initially for $n_{max} \gtrsim n_{max}^*$ the system jumps frequently between the two newly-formed potential wells, but eventually as $n_{max}$ is increased further the energetic barrier between them becomes larger making such jumps less likely.

# 6 Conclusion

We have investigated the steady state behaviour of a driven optomechanical system operating in the unresolved sideband regime ($\Omega_m, g_0 \ll \kappa$), where the oscillator displacement becomes a classical variable, even in the case of zero-temperature thermal and photonic environments. Here, we explored a special way of tuning the anharmonicity of the mechanical element, such that its effective potential may become purely quartic for certain parameters, at the transition before the system may become bistable. Adopting a Fokker-Planck description of the system, we presented numerical and analytical results that illustrate the measurable consequences of this anharmonicity. This is manifested by dramatic changes in the positions and line shapes of the spectral peaks for both the mechanics and the light, as well as the counter-intuitive reduction of the cavity photon population when the laser drive is increased close to/beyond the onset of the bistability, in analogy with the current blockade in electron transport.

## Acknowledgements

We acknowledge financial support from the French Agence Nationale de la Recherche (grant SINPHOCOM ANR-19-CE47-0012, and IMOON ANR-22- CE47-0015) and from the French government in the framework of the University of Bordeaux's France 2030 program / GPR LIGHT.

## A Optical forces

### A.1 Approximate solution for the cavity field

We repeat the equation of motion that determines the cavity field [Eq. (2) of the main text]

$$\dot{a} = \left[ i \left( \Delta + g_0 x / x_{\text{zpf}} \right) - \kappa/2 \right] \hat{a} - i\varepsilon + \sqrt{\kappa} \hat{a}_{\text{in}}. \tag{A.1}$$

We separate the dynamics of coherent and operator parts by substituting $\hat{a}(t) = \alpha(t) + \hat{\tilde{a}}(t)$, leading to the $c$-number equation $\dot{\alpha} = \left[ i \left( \Delta + g_0 x / x_{\text{zpf}} \right) - \kappa/2 \right] \alpha - i\varepsilon$. Splitting the displacement into a static and fluctuating part $x(t) = \bar{x} + \tilde{x}(t)$, we write the solution as

$$\alpha(t) = -i\varepsilon \int_0^t dt' e^{[i(\Delta + g_0 \bar{x}/x_{\text{zpf}}) - \kappa/2](t-t') + i[\Lambda(t) - \Lambda(t')]}, \tag{A.2}$$

where we introduced $\Lambda(t) = (g_0/x_{\text{xpf}}) \int_0^t dt' \tilde{x}(t')$, and we took $\alpha(t = 0) = 0$. Since $\kappa \gg \Omega_m$, we may expand the slow oscillator motion in the integrand according to $\Lambda(t') \approx \Lambda(t) + (g_0/x_{\text{zpf}}) \tilde{x}(t)(t'-t) + (g_0/x_{\text{zpf}}) \dot{\tilde{x}}(t)(t'-t)^2/2$, which when substituted yields the approximate expression

$$\alpha(t) \approx -i\varepsilon \int_0^t dt' e^{[i\Delta'[x(t)] - \kappa/2](t-t') - i(g_0/2x_{\text{zpf}}) \dot{\tilde{x}}(t)(t-t')^2}, \tag{A.3}$$

where we introduced the modified displacement $\Delta'[x(t)] = \Delta + g_0 x(t)/x_{\text{xpf}}$. Expanding the final term in the exponential and integrating gives for the coherent part of the cavity field:

$$\alpha(t) = \frac{i\varepsilon}{i\Delta'(x) - \kappa/2} + \frac{\varepsilon g_0 \dot{\tilde{x}}(t)/x_{\text{zpf}}}{(i\Delta'(x) - \kappa/2)^3}. \tag{A.4}$$

The corresponding operator equation is $\dot{\hat{a}} = [i\Delta'(x) - \kappa/2]\hat{a} + \sqrt{\kappa}\hat{a}_{\text{in}}$, whose approximate solution may be written via the same approach, finding:

$$\hat{a}(t) \approx \int_0^t dt' e^{[i\Delta'[x(t)] - \kappa/2](t - t')} \left[ 1 - \frac{i}{2} \frac{g_0}{x_{\text{zpf}}} \dot{x}(t)(t - t')^2 \right] \sqrt{\kappa}\hat{a}_{\text{in}}(t'). \qquad (A.5)$$

## A.2 Average optical forces

Armed with the solutions (A.4) and (A.5) the average radiation pressure force appearing on the right hand side of Eq. (5) of the main text may be written as $f = F_0(|\alpha|^2 + \langle\hat{a}^\dagger\hat{a}\rangle)$, where $F_0 = 2mx_{\text{zpf}}\Omega_m g_0$ is the force experienced by the oscillator due to one photon in the cavity. The operator term $\langle\hat{a}^\dagger\hat{a}\rangle \propto n_{\text{th}}^a$ is negligible since for optical cavity frequencies a reasonable experiment may satisfy the condition $n_{\text{th}}^a \ll 1$. We therefore have $f = f_0 - m\Gamma_{\text{opt}}(x)\dot{x}$, where $f_0 = 2mx_{\text{zpf}}\Omega_m g_0 n(x)$, with

$$n(x) = n_{\text{max}} \frac{\kappa^2/4}{\Delta'(x)^2 + \kappa^2/4}, \qquad (A.6)$$

$$\Gamma_{\text{opt}} = -4\Omega_m g_0^2 \Delta'(x)\kappa \frac{n(x)}{(\Delta'(x)^2 + \kappa^2/4)^2}, \qquad (A.7)$$

with the resonant cavity photon number $n_{\text{max}} = 4\varepsilon^2/\kappa^2$. The results for $f$ are the well-known cavity radiation-pressure force and the optical damping induced by the cavity field.

## A.3 Fluctuations of the optical force

We also characterise the quantum fluctuations of the optical force that give rise to the diffusion coefficient $D_{\text{opt}}$ appearing in the Fokker Planck equation (14). To do so, we require the correlation function $S_{ff}(t - t') = \langle\delta\hat{f}_0(t)\delta\hat{f}_0(t')\rangle$, where $\delta\hat{f}_0 = \hat{f}_0 - f_0$ in terms of the full force operator $\hat{f}_0 = 2mx_{\text{zpf}}\Omega_m g_0\hat{a}^\dagger\hat{a}$ and its average $f_0$. Here, we neglect the time-dependence of $x(t)$ since we expect optical force fluctuations to occur on a timescale fast compared to the oscillator motion, due to $\kappa \gg \Omega_m$. Plugging in the solution for $\hat{a} = \alpha + \hat{a}$ leads to the expression

$$S_{ff}(t - t') = F_0^2 \left[ \langle\hat{a}^\dagger(t)\hat{a}(t')\rangle\langle\hat{a}(t)\hat{a}^\dagger(t')\rangle + n(x)\left(\langle\hat{a}^\dagger(t)\hat{a}(t')\rangle + \langle\hat{a}(t)\hat{a}^\dagger(t')\rangle\right) \right]. \qquad (A.8)$$

The first term in square brackets gives a contribution at $t = t'$ that is proportional to $n_{\text{th}}^a \ll 1$ that we neglect in the following. We evaluate the second term using the input noise relations (4), finding the dominant contribution

$$S_{ff}(t - t') = F_0^2 n(x)e^{[i\Delta'(x) - \kappa/2](t - t')}, \qquad (A.9)$$

for $t > t'$, and the same expression with $\kappa \to -\kappa$ for $t < t'$. For the spectral noise weight we take the Fourier transform $S_{ff}(\Omega) = \int_{-\infty}^{\infty} dt e^{i\Omega(t - t')}S_{ff}(t - t')$, giving

$$S_{ff}(\Omega) = F_0^2 n(x)\frac{\kappa}{[\Omega + \Delta'(x)]^2 + \kappa^2/4}. \qquad (A.10)$$

Although the spectrum is not flat, we may approximate it as such since we shall be interested in oscillator behaviour at frequencies $\Omega \sim \Omega_m$, and focusing on parameters $\Delta' \sim \kappa$. We may therefore neglect $\Omega$ in the denominator and arrive at Eq. (15) of the main text, where we used the definition of the diffusion constant $S_{ff}(t - t') = D_{\text{opt}}\delta(t - t')$.

# B Displacement spectrum for the dissipationless anharmonic oscillator

## B.1 General formula

We include here the derivation of the displacement spectrum for an oscillator described by the anharmonic potential $V(x) = \mu x^2 + \nu x^4$ (with $\mu, \nu > 0$), described by a stationary probability distribution $P^{\text{st}}(x, p) = \mathcal{N} e^{-[p^2/2m + V(x)]/k_B T_{\text{eff}}}$. We have shown in Eq. (13) that the conservative dynamics of a mechanical oscillator of an optomechanical system in the unresolved sideband regime may be described by such a potential. From Eq. (13) the appropriate potential parameters are

$$\mu = m\bar{\Omega}_m^2/2, \tag{B.1a}$$

$$\nu = \frac{9\sqrt{3}}{16} \frac{n_{\max} \lambda^2 m^2 \Omega_m^4}{\hbar \kappa}, \tag{B.1b}$$

where $\bar{\Omega}_m = \Omega_m \sqrt{1 - n_{\max}/n_{\max}^*}$. We present the general solution in terms of $\mu$ and $\nu$, before applying it to our optomechanical context. The oscillator displacement fluctuations are described by the correlator $S_{xx}(t) = \int dx_0 \int dp_0 P^{\text{st}}(x_0, p_0) \tilde{x}_{x_0 p_0}(t) \tilde{x}_{x_0 p_0}(0)$, where $\tilde{x}(t) = x(t) - \langle x \rangle$ where $x(t)$ satisfies the equation $m\ddot{x} = F(x)$, and $(x_0, p_0)$ are the initial conditions. Since $x(t)$ is periodic, we may change the phase-space integration variables to integrate over a period for each possible closed trajectory, giving

$$S_{xx}(t) = \int_0^\infty dE \int_0^{T(E)} d\tau P^{\text{st}}(E) \tilde{x}_E(t + \tau) \tilde{x}_E(\tau), \tag{B.2}$$

where $T(E) = 2\pi/\omega(E)$ is the energy dependent oscillation period. We note that the presence of a constant term in $V(x)$ would simply change the lower limit of integration in $E$, but this constant offset does not affect the final result. Expanding $\tilde{x}_E(\tau)$ in its Fourier components as $\tilde{x}_E(\tau) = \sum_n e^{in\omega(E)\tau} \tilde{x}_n(E)$ allows to write the expression for the spectrum, given by $S_{xx}(\Omega) = \int_{-\infty}^\infty dt\, e^{i\Omega t} S_{xx}(t)$, as

$$S_{xx}(\Omega) = \int_{V_0}^\infty P^{\text{st}}(E) T(E) dE \sum_n 2\pi \delta(\Omega - n\omega(E)) \tilde{x}_n^2(E). \tag{B.3}$$

Introducing the oscillator energy levels which are solutions to $n\omega(E_n) = \Omega$ and using $\delta[\Omega - n\omega(E)] = \delta(E - E_n)/(n |\partial \omega(E)/\partial E|_{E_n})$, we arrive at

$$S_{xx}(\Omega) = \mathcal{N} (2\pi)^2 \sum_{n=0}^\infty \frac{e^{-E_n/k_B T_{\text{eff}}}}{n\omega(E_n) |\partial \omega(E)/\partial E|_{E_n}} \tilde{x}_n^2(E_n), \tag{B.4}$$

which is Eq. (19) of the main text. The problem of computing the spectrum is therefore reduced to finding $\omega(E)$ and $\tilde{x}_n(E)$.

Since we neglect dissipation, the energy is constant allowing to write $\dot{x} = \sqrt{2(E - V(x))/m}$. Integrating over a period leads to the expression for the period

$$T(E) = 4\sqrt{\frac{m}{2E}} x_{\max} \int_0^1 \frac{dz}{\sqrt{1 - \tilde{\mu} z^2 - \tilde{\nu} z^4}}, \tag{B.5}$$

where $x_{\max} = \sqrt{\mu(C(E) - 1)/2\nu}$ with $C(E) = \sqrt{1 + 4E\nu/\mu^2}$, $\tilde{\mu} = \mu x_{\max}^2/E$, and $\tilde{\nu} = \nu x_{\max}^4/E$. The integral appearing in Eq. (B.5) is the complete elliptic integral of the first kind, $K[-m(E)]$,

with the parameter $m(E) = (C-1)/(C+1)$ [49]. We therefore find for the energy-dependent oscillation frequency

$$\omega(E) = \frac{\pi}{2}\sqrt{\frac{\mu}{2}}\frac{\sqrt{C(E)+1}}{K[-m(E)]}. \tag{B.6}$$

For the calculation of the spectrum (19) we require its derivative, which is found to be (neglecting the $E$-dependence of $K[-m(E)]$):

$$\frac{\partial\omega(E)}{\partial E} = \frac{\pi}{2K[-m(E)]}\sqrt{\frac{\mu}{m(C(E)+1)}}\frac{\nu}{\mu^2 C(E)}. \tag{B.7}$$

The Fourier series coefficient is given by its corresponding definition

$$\tilde{x}_n(E) = \frac{1}{T(E)}\int_{-T/2}^{T/2}\tilde{x}_E(t)e^{-i2\pi nt/T(E)}dt. \tag{B.8}$$

We may in general re-express the integral over $t$ as an integral over $\tilde{x}_E$, via the relation $dt = d\tilde{x}_E/\sqrt{2(E-V(\tilde{x}_E))/m}$ and writing $t(\tilde{x}_E)$ in a way similar to Eq. (B.5), leading to the expression

$$\tilde{x}_n(E) = \frac{x_{\max}}{h(1)}\int_{-1}^{1}\frac{z}{\sqrt{1-\tilde{\mu}z^2-\tilde{\nu}z^4}}\cos\left[n\pi\frac{h(z)}{h(1)}\right]dz, \tag{B.9}$$

where we defined the integral function $h(z) = \int_{-1}^{z}dz/\sqrt{1-\tilde{\mu}z'^2-\tilde{\nu}z'^4}$, and used the symmetry relation $\tilde{x}_E(-t) = \tilde{x}_E(t)$ to eliminate the sinusoidal components. We notice from the above expression (B.9) that $\tilde{x}_n(E) \neq 0$ only for odd $n$, due to the symmetry relation: $\cos[n\pi h(-z)/h(1)] = (-1)^n\cos[n\pi h(z)/h(1)]$. The general integral (B.9) may be solved conveniently in some limiting cases.

## B.2 Weakly anharmonic case

We compute the required ingredients in Eq. (B.4) for the case of weak anharmonicity. From the general expression for the energy-dependent frequency (B.6) we approximate $\omega(E) \approx \omega(0) + \omega'(0)E$, and we use this expression to find the energy levels $E_n$. For the Fourier series coefficient, we take the harmonic expression: taking $\nu \to 0$ in Eq. (B.9) leads to finding the only nonzero coefficient $\tilde{x}_1(E) = -\sqrt{E/\mu}/2$. Computing the normalization factor $\mathcal{N}$ (again taking $\nu = 0$) and substituting everything into Eq. (B.4) leads to the total expression for the spectrum

$$S_{xx}(\Omega) = \pi\sqrt{\frac{1}{2m\mu}}\frac{1}{k_B T_{\text{eff}}}\frac{\Omega-\sqrt{2\mu/m}}{\omega|\omega'(0)|\omega'(0)}e^{-\frac{\Omega-\sqrt{2\mu/m}}{\omega'(0)k_B T_{\text{eff}}}}, \tag{B.10}$$

where anharmonicity (i.e. $\nu$) enters the above expression through $\omega'(0)$. Plugging in the optomechanical parameters from Eqs. (B.1) leads to the (adimensionalised) expression

$$\frac{S_{xx}(\Omega)}{x_{\text{zpf}}^2/\Omega_m} = 2\pi\frac{\hbar\Omega_m}{k_B T_{\text{eff}}}\frac{1}{\xi^2\tilde{\bar{\Omega}}_m}\frac{\tilde{\Omega}-\tilde{\bar{\Omega}}_m}{\tilde{\Omega}}e^{-\frac{\tilde{\Omega}-\tilde{\bar{\Omega}}_m}{\xi k_B T_{\text{eff}}/\hbar\Omega_m}}, \tag{B.11}$$

where $\xi = (9\sqrt{3}/8)n_{\max}\lambda^2/(\tilde{\bar{\Omega}}_m^3\tilde{\kappa})$, and tildes denote adimensionalisation of frequencies by $\Omega_m$. The spectrum (B.11) is peaked at $\Omega_{\max} = \bar{\Omega}_m + \xi k_B T_{\text{eff}}/\hbar$, with full width half maximum given by $\delta\Omega = \Delta_2\xi k_B T_{\text{eff}}/\hbar$, where $\Delta_2 \approx 2.446$ is the difference of the solutions of $xe^{1-x} = 1/2$. The result (B.11) indicates a line shape significantly different compared to the case of linearized optomechanics, where the spectral line shape remains Lorentzian with its width determined by the total dissipation [14].

### B.3   Purely quartic case

Taking $\mu = 0$ we compute the required ingredients for the spectrum (B.4), leading to the general expression

$$S_{xx}(\Omega) = \Upsilon \frac{\sqrt{m}}{(\nu k_B T_{\text{eff}})^{3/4}} \sum_{n=\{1,3,5,\dots\}}^{\infty} \frac{E_n \zeta_n^2}{n} e^{-E_n/k_B T_{\text{eff}}}, \tag{B.12}$$

with the constant $\Upsilon = 8\sqrt{2/\pi}(K[-1])^2/\Gamma[5/4]$, and $E_n = [\sqrt{2m}K[-1]\Omega/(\pi n)]^4/\nu$. We also introduced the numerical parameter $\zeta_n$ (as in Ref. [43]) coming from the Fourier series coefficient (B.9), which is given by

$$\zeta_n = \int_{-1}^{1} \frac{z}{g(1)\sqrt{1-z^4}} \cos\left[n\pi \frac{g(z)}{g(1)}\right] dz, \tag{B.13}$$

which involves the integral equation $g(z) = \int_{-1}^{z} dz'/\sqrt{1-z'^4}$, and $g(1) = 2K[-1]$. Evaluating numerically we have that $\zeta_1 = -0.478$, $\zeta_3 = -0.022$ and $\zeta_5 = -9.3 \times 10^{-4}$, and so the main contribution to Eq. (B.12) is from the $n = 1$ first harmonic. Retaining only this term and plugging in the optomechanics value of $\nu$ from Eq. (B.1b) leads to the approximate expression

$$\frac{S_{xx}(\Omega)}{x_{\text{zpf}}^2/\Omega_m} = \Xi \left(\frac{\tilde{\kappa}}{\lambda}\right)^{7/4} \left(\frac{\hbar\Omega_m}{k_B T_{\text{eff}}}\right)^{3/4} \tilde{\Omega}^4 e^{-\eta\tilde{\Omega}^4}, \tag{B.14}$$

with constant $\Xi = (96\sqrt{2}/\pi^{9/2})(4/3)^{3/4}((K[-1])^6/\Gamma[5/4])\zeta_1^2$, and parameter $\eta = (16/3\pi^4)[(K[-1])^4/\lambda]\hbar\kappa/k_B T_{\text{eff}}$. The expression (B.14) has a peak at $\Omega_{\text{max}} = \eta^{-1/4}\Omega_m$ and a full width half maximum $\delta\Omega = \Delta_4\Omega_{\text{max}}$, where $\Delta_4 \approx 0.585$ is the difference between the solutions of $x^4 e^{1-x^4} = 1/2$. This expression (B.14) is plotted as the dashed blue line in Fig. 3(d), and dividing by its maximum gives the normalised formula given in Eq. (20) of the main text.

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
