# Peer review of "Tunable anharmonicity in cavity optomechanics in the unresolved sideband regime"

_SciPost Physics_

## Round 1 · Referee Report · Anonymous (Referee 1) · 2025-4-23

Strengths

  1. Very clear paper
  2. Original physics
  3. Interesting topic
  4. Experimentally relevant physics
  5. Good physical interpretations
  6. Already in suitable shape for publication

Report

"Tunable anharmonicity in cavity optomechanics in the unresolved sideband regime"

Although this paper deals with a rather standard optomechanical system in a particular regime, this analysis is new to my knowledge. I found the paper very well written, easy to read, well presented, with an adequate physical discussion matching each of the theoretical predictions.

The paper studies the steady states of a mechanical oscillator and an electromagnetic cavity when in optomechanical coupling with a high single-photon parametric cooperativity. This is somewhat unusual for the optomechanics community as the competition has been a lot towards reaching high *enhanced* cooperativities. From what I understand, in both cases, the single-photon coupling rate g0 should be as high as possible, but high enhanced cooperativities simultaneously require small mechanical decay rates, which are intrinsically difficult to obtain in well coupled systems, while high single-photon parametric cooperativities ask for small frequencies, which is easy and typically allows for larger g0. The authors show that regimes where the coupling potential is quartic in the drive detuning, and even purely quartic, rather than quadratic, are achievable at relatively low cavity photon number (~50-100). I make the last remark in comparison (still) to studies aiming at the largest enhanced cooperativities which attempt to counterbalance the smallness of g0 by using very large optical/microwave powers driving tens of thousands or millions of photons into the cavity. The reason why this is notable is not that the two types of systems should compete as they intrinsically present different interesting physics, but because, in such systems with intrinsically large single-photon parametric cooperativity as the one cited in Ref [34], the signatures described by the authors (a flattening of the thermal spectra, transitions between one-well and two-wells potentials, significant pulls of the cavity frequency, sidebands in the cavity emission spectrum) seem easily within experimental reach with low optical powers. In other words, this work strikes me not only for its clarity but for its experimental relevance.

I am reduced to listing typos to make this report any useful. I can already recommend publication at this stage, without a need for a second review.

Typos:

Page 5, last paragraph: "oscillitor"

Page 12 "Although some optomechanics experimental setups *give access allow* direct measurement
of the mechanical displacement"

Page 14, next-to-last paragraph, "principle spectral peak" -> principal

Recommendation

Publish (surpasses expectations and criteria for this Journal; among top 10%)

  • validity: top
  • significance: high
  • originality: high
  • clarity: top
  • formatting: perfect
  • grammar: excellent

Author:  Jonathan Wise  on 2025-05-22  [id 5507]

(in reply to Report 1 on 2025-04-23)

We are very grateful to the Referee for reviewing our manuscript, for their positive comments, and for highlighting the experimental relevance of our results.

We thank the Referee for identifying these typos. We corrected them in the text.

---

## Round 1 · Referee Report · Anonymous (Referee 2) · 2025-4-28

Strengths

1- Interesting and relevant topic 2- Thorough and comprehensive analysis of the problem 3- Realistic connection with experimentally accessible parameter ranges

Weaknesses

Several points and approximations are unclear.

Report

This work is interesting and valuable. I am inclined to recommend it for publication; however, I have some technical questions and invite the authors to clarify several issues that are unclear in the current version.

1) In Appendix A.1, when going from Eq. (A.3) to Eq. (A.4), the authors perform an expansion of the exponential term proportional to the velocity. However, the validity and physical meaning of this expansion are not discussed. Could the authors clarify under what assumptions this expansion is justified?

2) In Eq. (A.5), for the fluctuating part of the cavity field, the term proportional to the velocity is completely neglected (i.e., the velocity of the oscillator does not appear in the fluctuations of the optical force), whereas in Eq. (A.4) the linearized term proportional to the velocity is retained (and is indeed responsible for the optical damping). Could the authors explain this asymmetry in the treatment of the velocity terms?

3) Starting from Eq. (5), the Langevin equation, the authors derive the Fokker-Planck equation (Eq. (14)). Unlike the simple Brownian motion case — where the damping coefficient and the force correlator are constants — here the optical damping (Eq. (7)) and the fluctuation force (Eq. (A.10)) explicitly depend on the position x. It is therefore not obvious that one can simply obtain Eq. (14) from Eq. (5) by replacing the damping and diffusion coefficients with position-dependent functions. The formal derivation should involve the Kramers–Moyal expansion (see Ref. [40]). Could the authors provide a discussion of this point?

4) In Eqs. (16) and (17), the authors introduce the concept of an "effective temperature." However, it is not clear how this definition is justified, given that D_tot (or better D_opt, the optical diffusion coefficient) and the effective detuning are position-dependent functions, see Eq. 7 and Eq. 15. Could the authors clarify this point?

5) The results shown in Figure 4 are very interesting. Would it be possible to include in the inset a cut at n_max /n_max∗ larger than one?

6) In Figure 4, the “second harmonic” is also shown. How was this curve obtained — numerically or through an approximate analytical formula? Could the authors provide details?

7) In Figure 5, the average cavity occupation displays steep jumps. However, from Eq. (23), the integral involves the product of Pst and a Lorentzian, both of which are smooth functions (see also Fig. 3 for Pst ). Since only the marginal distribution in x is needed to calculate this quantity, a smooth behavior would be expected. Could the authors clarify this apparent discrepancy?

Requested changes

See list above.

Recommendation

Ask for major revision

  • validity: high
  • significance: high
  • originality: good
  • clarity: good
  • formatting: good
  • grammar: excellent

Author:  Jonathan Wise  on 2025-05-22  [id 5508]

(in reply to Report 2 on 2025-04-28)

The Referee writes:

This work is interesting and valuable. I am inclined to recommend it for publication; however, I have some technical questions and invite the authors to clarify several issues that are unclear in the current version.

Our response: We are grateful to the Referee for reviewing our manuscript and for their positive comments. We address the Referee's questions below. All modifications mentioned figure in the revised manuscript resubmitted for publication.

The Referee writes:

1) In Appendix A.1, when going from Eq. (A.3) to Eq. (A.4), the authors perform an expansion of the exponential term proportional to the velocity. However, the validity and physical meaning of this expansion are not discussed. Could the authors clarify under what assumptions this expansion is justified ?

Our response: We thank the Referee for their suggestion to improve the clarity of the manuscript. The expansion in Eq. (A.3) is mainly based on the separation of timescales between the fast cavity dynamics and the slow mechanical dynamics ($\kappa \gg \Omega_{\rm m}$), which is a central assumption throughout the manuscript. In the revised version of Appendix A.1, we now provide a detailed derivation and explicitly state each approximation involved.

The Referee writes:

2) In Eq. (A.5), for the fluctuating part of the cavity field, the term proportional to the velocity is completely neglected (i.e., the velocity of the oscillator does not appear in the fluctuations of the optical force), whereas in Eq. (A.4) the linearized term proportional to the velocity is retained (and is indeed responsible for the optical damping). Could the authors explain this asymmetry in the treatment of the velocity terms?

Our response: The apparent asymmetry in the treatment of these two velocity dependent terms is due to the difference in their prefactors - the term in (A.4) is enhanced by the average cavity occupation, while that of (A.5) is proportional to the occupation $n^a_{\rm th}$ of the electromagnetic environment. In the manuscript, we assume that the optical field has zero thermal occupation $n^a_{\rm th}\simeq0$, or equivalently $k_{\rm B} T/\hbar\omega_{\rm c} \simeq 0$ (see Appendix A.2). This approximation is valid even at room temperature for optical frequencies, such as $\omega_{\rm c}$, and is routinely applied in cavity optomechanics. We modified Appendix A.2 to clarify this asymmetry in the treatment of the velocity.

The Referee writes:

3) Starting from Eq. (5), the Langevin equation, the authors derive the Fokker-Planck equation (Eq. (14)). Unlike the simple Brownian motion case — where the damping coefficient and the force correlator are constants — here the optical damping (Eq. (7)) and the fluctuation force (Eq. (A.10)) explicitly depend on the position $x$. It is therefore not obvious that one can simply obtain Eq. (14) from Eq. (5) by replacing the damping and diffusion coefficients with position-dependent functions. The formal derivation should involve the Kramers–Moyal expansion (see Ref. [40]). Could the authors provide a discussion of this point?

Our response: We agree that standard formal derivation of the Fokker-Planck equation exploit the Kramers-Moyal expansion truncated at second order. As one can readily verify, for instance from Eq. (4.14) in Ref.[40], a position dependence of the momenta does not change the result, and is perfectly compatible with the derivation. The main assumptions involved in the derivation are that the stochastic force is (i) delta-correlated and (ii) Gaussian-distributed. In our analysis, the condition $\Omega_{m} \ll \kappa$ ensures that the optical force satisfies both assumptions, thus justifying the use of the standard Fokker-Planck formalism.

Assumption (i): In Appendix A.3 of our manuscript, we analyze the fluctuations of the optical force acting on the mechanical system. We show that, in the limit $\Omega_{m} \ll \kappa$, the noise spectrum is approximately flat near the mechanical resonance frequency, which is the frequency range of interest. This validates the approximation that the optical force is effectively delta-correlated in time.

Assumption (ii): Furthermore, the fluctuations of the optical force occur with a typical timescale $\Delta t = 2\pi/\kappa$. Since $\Omega_{m} \ll \kappa$, many such fluctuations occur within one mechanical oscillation period. This implies that the force experienced by the mechanical oscillator results from the sum of many independent stochastic contributions. By the central limit theorem, these fluctuations are expected to be Gaussian-distributed.

Thus, we believe that the Fokker-Planck equation presented in the manuscript is valid in the regime considered. We added a sentence before the Fokker-Planck equation (14) to specify the validity conditions and refer to App. A.3, where we now summarize the discussion above.

The Referee writes:

4) In Eqs. (16) and (17), the authors introduce the concept of an ``effective temperature." However, it is not clear how this definition is justified, given that $D_{tot}$ (or better $D_{opt}$, the optical diffusion coefficient) and the effective detuning are position-dependent functions, see Eq. 7 and Eq. 15. Could the authors clarify this point?

Our response: We thank the Referee for pointing this out. They are totally right. For the equilibrium condition to be valid, the position dependence of the coefficients has to be negligible within the fluctuation range of the probability. We detail this point in the revised version in Sec. 3.3. The condition is $\lambda=2g_0^2/(\Omega_{ m}\kappa)\ll1$, that we assume in the numerics of the paper.

The referee writes:

5) The results shown in Figure 4 are very interesting. Would it be possible to include in the inset a cut at $n_{max}/n_{max}^*$ larger than one?

Our response: We thank the Referee for the positive feedback. As suggested, we have included an additional curve in the inset corresponding to the bistable regime. We modified the figure caption accordingly.

The Referee writes:

6) In Figure 4, the “second harmonic” is also shown. How was this curve obtained — numerically or through an approximate analytical formula? Could the authors provide details?

Our response: The analytical expression for the first harmonic is given in the text, and the second harmonic is simply twice this expression. We made more explicit in the figure caption and the main text that the yellow curves in Fig. 4 correspond to analytical expressions.

The Referee writes:

7) In Figure 5, the average cavity occupation displays steep jumps. However, from Eq. (23), the integral involves the product of Pst and a Lorentzian, both of which are smooth functions (see also Fig. 3 for Pst). Since only the marginal distribution in x is needed to calculate this quantity, a smooth behavior would be expected. Could the authors clarify this apparent discrepancy?

Our response: We thank the Referee for raising this important point. We begin by noting that the average of any observable, say $A(x,p, \Delta)$ [where with $\Delta$ we indicate all other parameters], is computed via an integral involving the distribution $P(x,p)$ such that $\bar A(\Delta) = \int dx dp \,P(x,p) A(x,p, \Delta)$. The fact that $A$ and $P$ are both smooth functions of $x$ and $p$ is not sufficient to guarantee that the average $\bar{A}$ is a smooth function of $\Delta$. Typically $A$ is a smooth function of $\Delta$. However, the same is not necessarily true of the distribution $P$, since it is obtained by solving a nonlinear equation, which may undergo sudden changes - i.e. bifurcations - as the parameters are varied. These bifurcations can result in abrupt changes in observable quantities, despite the smoothness of the probability distribution for fixed parameters. We clarified this issue in the end of Sec. 5.1. in the revised version of the manuscript.

Attachment:

SciPostPhys_Ref2.pdf

---

## Editorial Decision

resubmitted